# Small Intestinal Contrast Ultrasonography (SICUS) in Crohn’s Disease: Systematic Review and Meta-Analysis

**DOI:** 10.3390/jcm12247714

**Published:** 2023-12-15

**Authors:** Giuseppe Losurdo, Margherita De Bellis, Raffaella Rima, Chiara Maria Palmisano, Paola Dell’Aquila, Andrea Iannone, Enzo Ierardi, Alfredo Di Leo, Mariabeatrice Principi

**Affiliations:** 1Section of Gastroenterology, Department of Precision and Regenerative Medicine and Ionian Area, University of Bari, 70124 Bari, Italy; margideb@gmail.com (M.D.B.); paoladellaquila@tiscali.it (P.D.); ianan@hotmail.it (A.I.); ierardi.enzo@gmail.com (E.I.); alfredo.dileo@uniba.it (A.D.L.); b.principi@gmail.com (M.P.); 2Internal Medicine Unit “C. Frugoni”, Department of Interdisciplinary Medicine, University of Bari “Aldo Moro”, Piazza Giulio Cesare 11, 70124 Bari, Italy; chiarapalmi8@gmail.com

**Keywords:** Crohn’s disease, diagnosis, ultrasound, magnetic resonance enterography, oral contrast, SICUS

## Abstract

The diagnosis of Crohn’s Disease (CD) is based on a combination of clinical symptoms, laboratory tests, endoscopy, and imaging data. In Small Intestine Contrast Ultrasonography (SICUS), the ingestion of a macrogol solution as an oral contrast medium may optimize image quality. We performed a meta-analysis to evaluate the diagnostic performance of SICUS for CD. A literature search was performed in August 2023. We selected only studies where SICUS was compared to a technique that allows the assessment of the whole gastrointestinal tract, such as an MRE, a CT scan, or a surgical evaluation. We estimated pooled weighted sensitivity, specificity, and likelihood ratio for positive and negative tests (PLR/NLR) of SICUS. Summary receiver operating characteristic curves (SROC) were drawn, and pooled areas under the curve (AUC) were calculated. Five studies with 325 CD patients were included. SICUS showed a pooled sensitivity for the diagnosis of 95% (95% confidence interval CI 89–99%), a specificity = 77% (95% CI 60–90%), and the AUC was 0.94. SICUS demonstrated a pooled sensitivity for strictures of 78% (95% CI 63–88%) and a specificity = 96% (95% CI 85–99%), with AUC = 0.93. For abscesses, SICUS demonstrated a pooled sensitivity of 100% (95% CI 59–100%) and a specificity of 90% (95% CI 74–98%). Fistulae were detected with a pooled sensitivity of 77% (95% CI 46–95%) and a specificity of 92% (95% CI 75–99%). SICUS demonstrated excellent diagnostic performance compared to the gold standard despite some clinical scenarios (stenosis/fistulae) showing suboptimal diagnostic effectiveness.

## 1. Introduction

Crohn’s disease (CD) is a chronic condition with immunological pathogenesis, which can affect any site of the gastrointestinal tract in a segmental and transmural way, from the mouth to the anus. A total of 50% of patients have an involvement of the terminal ileum and colon, while 30% have an isolated small bowel involvement, while the remaining 20% of cases are confined to the colon. Among patients with small bowel disease, the terminal ileum is affected in 90% of cases [1].

CD is characterized by periods of remission alternated with phases of a flare-up. The inflammatory process can evolve towards either a fibrostenotic-obstructive picture or a penetrating-fistulizing one [1]. Symptoms can be insidious or nonspecific and depend on the site and severity of the disease. The development of adhesions leads to the formation of fistulas, as CD induces transmural damage. Abdominal and pelvic abscesses develop in 10 to 30% of patients. Other complications include intestinal obstruction, occurring in 40% of cases, massive hemorrhage, malabsorption, and severe perianal disease [1].

The diagnosis of CD relies on a combination of clinical symptoms, laboratory tests, and imaging data [2]. Ileopancolonoscopy is the first technique involved in diagnosis, management, and monitoring; however, endoscopy is not always a thorough investigation and is limited by invasiveness, poor patient compliance, and a risk of bowel perforation. Ileopancolonoscopy also fails to evaluate the extent of ileal disease, transmural damage, and lesions in the perineal region, such as fistulas and abscesses. Therefore, other imaging techniques, including ultrasonography, computed tomography (CT), and magnetic resonance enterography (MRE), have been more frequently used recently. Indeed, transabdominal ultrasound is non-invasive, does not use ionizing radiation, and is easily accepted by patients. Nowadays, intestinal ultrasound use is increasing as a clinically important first-line technique both in patients with suspected CD and disorder follow-up [3].

In particular, in small intestinal contrast ultrasonography (SICUS), introducing an oral contrast medium may optimize image quality and increase sensitivity and diagnostic accuracy in detecting small intestine lesions [3]. This method, therefore, has become relevant for investigating patients with CD for the classification of disease activity, the analysis of small bowel stenosis/mural fibrosis, and the evaluation of specific therapy responses [3]. In SICUS, patients are examined in the fasting state and after ingestion of an oral macrogol contrast solution consisting of polyethylene glycol (PEG) in powder at a dose ranging from 125 to 800 mL (usually 375 mL), dissolved in 250 mL of water [3]. The introduction of an oral contrast medium allows for distension of the intestinal lumen, with better visualization of the intestinal wall and accuracy in detecting complications related to CD, including strictures, abscesses, and fistulas [3]. Furthermore, its use has been proposed in the preoperative evaluation of CD, offering precision in detecting the presence of dilatation upstream of the stenosis [4,5]. Thus, SICUS has emerged as a valuable, well accepted, and radiation-free technique in the detection of intestinal damage in CD.

Therefore, we aimed to perform a systematic review and meta-analysis in order to evaluate the pooled diagnostic performance of SICUS in patients with CD in comparison to gold-standard techniques able to assess the transmural activity of the disease.

## 2. Materials and Methods

### 2.1. Eligibility Criteria and Study Selection

Methods of analysis and inclusion criteria were based on “Preferred Reporting Items for Systematic Reviews and Meta-Analyses” (PRISMA) recommendations [6], and its extension for diagnostic test accuracy (PRISMA-DTA) was taken into account [7]. A PRISMA-DTA checklist is provided in Appendix A. We excluded review articles, experimental in vitro studies, and single case reports. In cases of studies analyzing overlapping periods from the same registry/database, we considered only the study that examined the longest period and the largest number of patients.

### 2.2. Data Collection Process

A literature search was performed and updated in August 2023. Relevant publications were identified through research in PubMed, Web of Science, and Scopus. Only in extenso papers were selected; therefore, abstracts or conference proceedings were excluded. The search terms were Crohn’s Disease, ultrasound, oral contrast, and SICUS. We used the following string, using Boolean operators AND/OR: Crohn’s Disease AND (ultrasound OR small bowel OR oral contrast OR SICUS). We selected only studies in which SICUS was compared to a technique that allowed assessment of the whole gastrointestinal tract, such as magnetic resonance enterography (MRE), computed tomography (CT), or surgical evaluation. Therefore, in the case of comparison with colonoscopy, the study was excluded. In our search strategy, we included only papers in which the gold standard was able to assess all the bowel walls; therefore, if SICUS was compared only to enteroclysis or capsule endoscopy, it could not be included. Titles and abstracts of papers were screened by two reviewers (MDB and RR). Successively, data were extracted from the relevant studies by one reviewer and checked by a second reviewer, and thus inserted into dedicated tables. A third reviewer (GL) came to a decision on any disagreements.

Reviewers independently extracted from each paper the following data: (i) publication year, (ii) country, (iii) single- or multi-center study, (iv) study design, (v) number of patients included, (vi) oral contrast agent, (vii) ultrasound device, and (viii) number of true positive/negative and false positive/negative results. If the study did not provide sufficient data to extract true positive/negative and false positive/negative outcomes, it was excluded from the final analysis.

### 2.3. Summary Measures and Planned Methods of Analysis

The end-point was to estimate the pooled weighted sensitivity, specificity, and likelihood ratio for positive and negative tests (PLR and NLR, respectively) and the diagnostic odd ratio (DOR) of SICUS. Summary receiver operating characteristic curves (SROC) were drawn, and pooled areas under the curve (AUC) were calculated. A random effect model was followed in all analyses. We assessed heterogeneity using the χ^2^ test, and if it was statistically significant, the I^2^ statistic was computed. If necessary, a subgroup analysis was performed. The data were expressed as proportions/percentages, and 95% confidence intervals (CI) were calculated. A *p*-value < 0.05 was considered statistically significant. All analyses were performed according to the general principles of meta-analysis [8]. The MetaDisc software version 1.4 was used [9].

Two reviewers (GL and PD) independently assessed the quality of the included studies using the Quality Assessment of Diagnostic Accuracy Studies version 2 (QUADAS-2) instrument [10]. This tool is designed to assess the quality of primary diagnostic accuracy studies for inclusion in the systematic review (Appendix A).

## 3. Results

### 3.1. Study Selection

After a bibliography search, five studies were included in the analysis [11,12,13,14,15]. Such studies are reported in Table 1. The process of study selection is summarized in Figure 1. All studies but one [12] were performed in the adult population. Three studies were performed in the UK [11,12,13] and two in Italy [14,15]. All studies used PEG as an oral contrast agent, with final volume ranging from 250 mL to 1000 mL. Overall, 325 patients with CD were recruited.

### 3.2. Final Diagnosis

The term “final diagnosis” in the selected papers was defined as “the final judgement of the physician after performing all diagnostic tests and referred only to CD”. For this analysis, four studies provided sufficient data [11,12,13,15]. One hundred and twenty-one patients were recruited. SICUS showed a pooled sensitivity of 95% (95% CI 89–99%), a specificity of 77% (95% CI 60–90%), a positive LR of 2.73 (0.93–8.05), a negative LR of 0.15 (0.06–0.41) and a DOR = 24.94 (5.90–105.47). The AUC of the SROC curve was 0.94. Further details are shown in Figure 2.

### 3.3. Strictures

The presence of strictures was examined in 94 patients within all studies. SICUS demonstrated a pooled sensitivity of 78% (95% CI 63–88%), a specificity of 96% (95% CI 85–99%), a positive LR of 6.37 (0.93–8.05), a negative LR of 0.23 (0.05–1.15), and a DOR = 30.99 (7.07–135.82). The AUC of the SROC curve was 0.93. Such results are summarized in the plots in Figure 3.

### 3.4. Abscesses

This analysis was possible for only forty patients overall in three studies [13,14,15]. SICUS demonstrated a pooled sensitivity of 100% (95% CI 59–100%), a specificity of 90% (95% CI 74–98%), a positive LR of 6.34 (1.94–21.17), a negative LR of 0.13 (0.02–0.84), and a DOR = 53.08 (5.07–555.11). It was not possible to calculate the AUC of SROC. Such results are reported in the plots of Figure 4. Of note, since one study provided several zero values, the corresponding diamond in the plots was not drawn in the figures.

### 3.5. Fistulae

This outcome was evaluated in 55 patients overall across three studies [13,14,15]. SICUS showed a pooled sensitivity of 77% (95% CI 46–95%), a specificity of 92% (95% CI 75–99%), a positive LR of 8.82 (2.23–34.88), a negative LR of 0.29 (0.08–1.11), and a DOR = 33.75 (3.13–363.90). It was not possible to calculate the AUC of SROC. Such results are reported in the plots of Figure 5. Of note, since one study provided several zero values, the corresponding diamond in the plots was not drawn in the figures.

### 3.6. Dilation

Three studies [12,13,14] investigated the presence of pre-stenotic luminal dilation in 61 patients. We found a pooled sensitivity of 100% (95% CI 80–100%), a specificity of 80% (95% CI 65–90%), a positive LR of 3.99 (2.30–6.94), a negative LR of 0.10 (0.02–0.49) and a DOR of 41.03 (6.62–254.27). The AUC of the SROC curve was 0.91. Such results are summarized in the plots of Figure 6.

## 4. Discussion

Since CD may affect every segment of the digestive system, with transmural involvement, endoscopy techniques are not always adequate for the investigation of the whole bowel length. Imaging procedures with panoramic spatial resolution are necessary to integrate clinical and endoscopic. MRE is considered the gold standard nowadays, but it has some drawbacks. For example, it is not available in all centers, and it is time consuming. CT enterography has emerged as an alternative as it is more widespread and more rapid, despite radiation exposure being a relevant limit. SICUS is an ultrasound-based method that explores bowel loops and is able to identify wall thickness, intestinal motility, perfusion using a Doppler scan, and possible complications such as stenosis, dilation, fistulae, and abscesses [16]. Oral ingestion of a contrast (usually PEG dissolved in a volume of water ranging from 250 to 1000 mL) may help to increase the sensitivity of ultrasound since it may enhance some characteristics such as pre-stenotic dilation; furthermore, lumen distension is useful to better evaluate the thickness of the wall and the feature of its layers. Another advantage of SICUS is its dynamic peculiarity, which allows one to focus on a detail and analyze it under several planes and direct motion. Conversely, artifacts, interposition of air and loops, and power of resolution might be intrinsic limitations of this method. Bowel ultrasound is a highly acceptable and well-tolerated tool for monitoring disease activity in IBD patients [17].

SICUS is a safe technique: side effects were described only in the articles by Onali and Pallotta, and no side effects were recorded in any patients in these two studies. Nevertheless, it could be argued that in case of clinically significant strictures, a liquid overload may elicit subocclusion symptoms; therefore, particular care should be taken into account.

The first relevant finding of our meta-analysis was a very high sensitivity (95%) for CD diagnosis, while the specificity was slightly lower (77%). This could be justified by the use of a cut-off value of 3 mm for bowel wall thickness in most studies [11,14,15]. Indeed, some Authors have proposed higher cut-offs of 4 mm [18] or 5 mm [19] to increase specificity. On the other hand, we found a sensitivity of 78% for stenosis detection. This finding is in agreement with previous studies, showing that the sensitivity of ultrasound for stenosis detection may range around 80% [20]; therefore, some stenotic areas may have been missed at SICUS. Apart from such results, most studies confirmed that SICUS has a good agreement with gold standard procedures. The perspective is different for fistulae. The ultrasound pattern evocating a possible fistula may not be univocal, and several possible signs have been proposed, often related to the experience of the examiner [21]. Indeed, some further evidence showed a sensitivity close to 70% for detecting fistulae, even when other ancillary techniques, such as water immersion, were adopted [22]. However, results are conflicting in the literature, as an additional study showed a sensitivity of 100% and a specificity of 98% [23], thus underlining that the expertise of the observer may be a main issue and a bias to be highlighted when reporting results in a systematic review. Rectum may be difficult to examine by ultrasound, and this may explain why some cases of fistulae or abscesses may be missed by SICUS [24]. A different approach in this case, i.e., trans-perineal ultrasound examination, could add sensitivity to SICUS, as shown in some evidence from literature [25], provided that the operator has a sufficient level of expertise when scanning the perineal and perirectal areas.

In pediatric populations, sensitivity and specificity were even higher, >90% [26,27]. This could be explained as CD commonly affects the small bowel in children [28]. In this regard, the pediatric population could be the ideal target for SICUS due to non-invasiveness and lack of radiation exposure. A recent expert consensus clearly underlined this point and promoted the standardization of the technique, as basic equipment requirements, patient selection, preparation and positioning, technical considerations, and limitations may cause a lack of reproducibility among operators [29]. In this regard, the Simple Pediatric Activity Ultrasound Score has been published, demonstrating a substantial agreement between ultrasound and endoscopy for all disease locations (weighted k = 0.85) and substantial agreement for ileocolonic disease (weighted k = 0.96) [30]. In children, a bowel wall thickness superior to 1.9 mm had a sensitivity of 64%, a specificity of 76%, and an area under the curve of 0.743 for detecting inflammation, compared to ileo-colonoscopy [31].

A previous meta-analysis from 2016 comparing imaging and endoscopy has already been published on the topic [32], showing a pooled sensitivity of 88.3% and specificity of 86.1%. However, such meta-analysis was enclosed as the gold standard for both imaging techniques and endoscopy; therefore, it was hampered by a relevant heterogeneity. Moreover, comparing ileopancolonoscopy and SICUS is possible only for colonic or ileal disease; therefore, the remaining segments of the small bowel cannot be taken into account. In the present analysis, we recruited only studies in which the gold standard could provide a panoramic and transmural evaluation of the whole intestine, thus providing a more homogeneous and punctual comparison. The comparison between colonoscopy and SICUS may be useful only for assessing the activity of terminal ileum or to predict disease reactivation after ileo-cecal resection; in this regard, another meta-analysis confirmed a very good sensitivity (99%) and specificity (74%) to detect post-surgical recurrence [8]. 

Despite the accuracy of the methodological approach, the current meta-analysis has some limitations. The most important one is that for some outcomes, the total amount of patients analyzed is low (about 50 for fistulae and abscesses). Another limitation is that we were not able to sub-analyze the localization of the disease (proximal or distal small bowel) because such data were not available in all the studies included. 

In conclusion, our meta-analysis confirmed that SICUS has a very good performance compared to the gold standard as well as the evident advantage of its easy availability and feasibility. Some disadvantages might be the level of operator experience and the risk of missing some pictures. For these reasons, the present study supports SICUS’s usefulness for the periodic monitoring of CD evolution; nevertheless, a panoramic MRE should be performed upon initial classification and in the event of significant progression of the disease. Some authors have suggested that the diagnostic accuracy of intestinal ultrasound could be optimized using contrast-enhanced ultrasonography [33], and this might be an additional step to reach an effectiveness comparable to current imaging gold standards. 

## Figures and Tables

**Figure 1 jcm-12-07714-f001:**
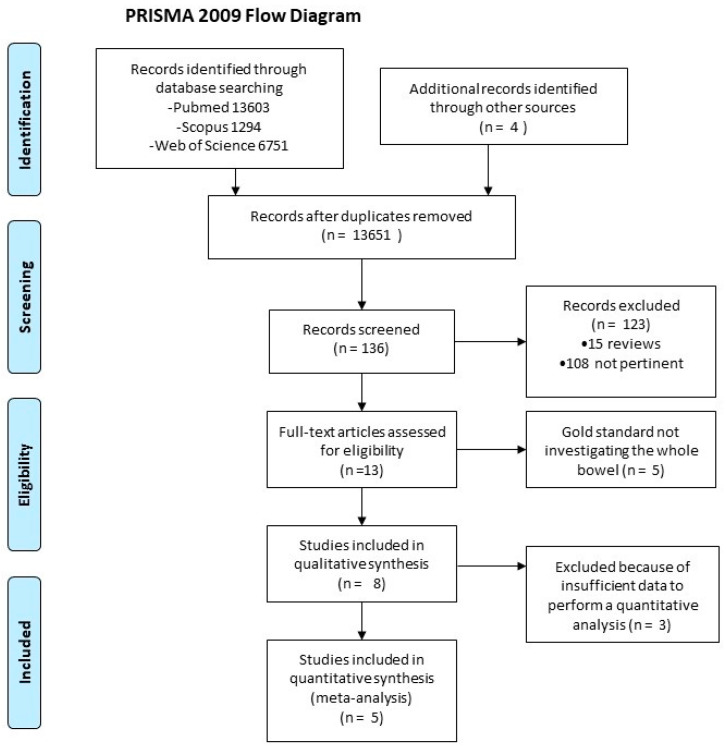
Flowchart summarizing the process of figure selection.

**Figure 2 jcm-12-07714-f002:**
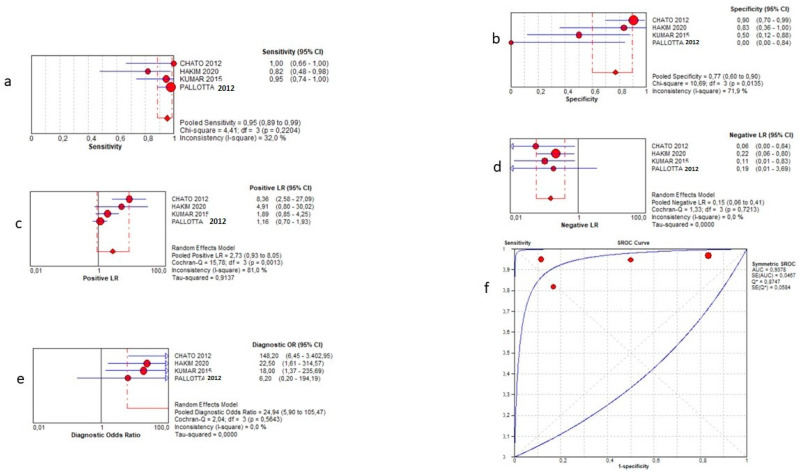
Sensitivity (**a**), specificity (**b**), positive likelihood ratio (**c**), negative likelihood ratio (**d**), diagnostic odd ratio (**e**), and AUC (**f**) for diagnosis of CD.

**Figure 3 jcm-12-07714-f003:**
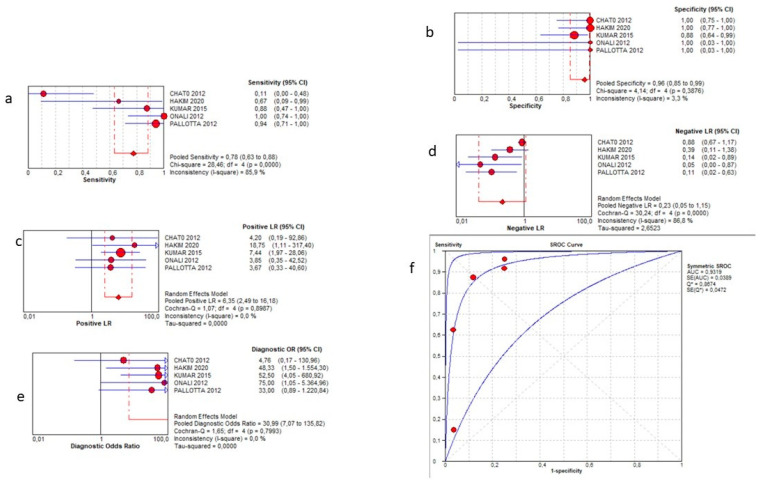
Sensitivity (**a**), specificity (**b**), positive likelihood ratio (**c**), negative likelihood ratio (**d**), diagnostic odd ratio (**e**), and AUC (**f**) for stenosis detection.

**Figure 4 jcm-12-07714-f004:**
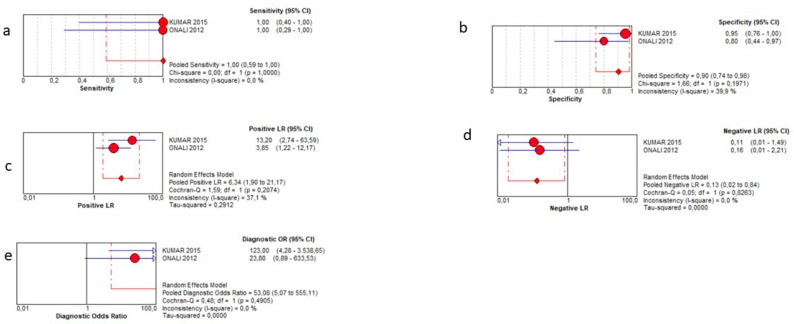
Sensitivity (**a**), specificity (**b**), positive likelihood ratio (**c**), negative likelihood ratio (**d**), and diagnostic odd ratio (**e**) for abscess detection.

**Figure 5 jcm-12-07714-f005:**
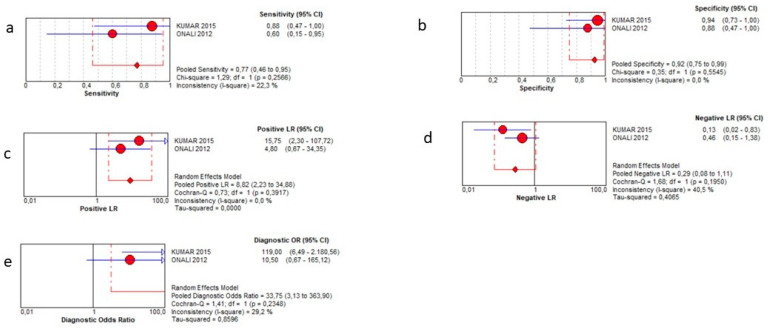
Sensitivity (**a**), specificity (**b**), positive likelihood ratio (**c**), negative likelihood ratio (**d**), and diagnostic odd ratio (**e**) for fistulae detection.

**Figure 6 jcm-12-07714-f006:**
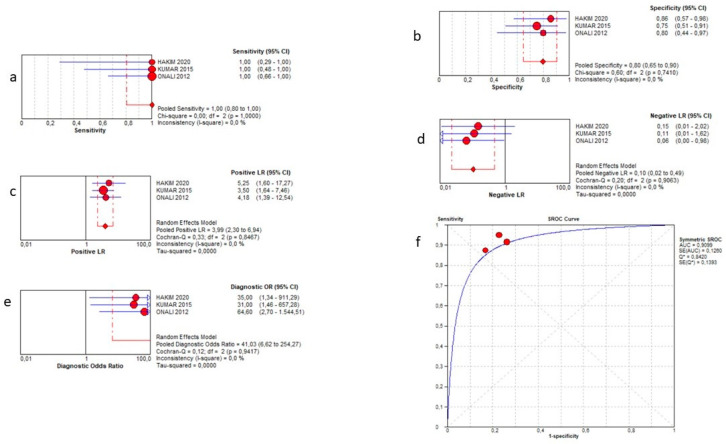
Sensitivity (**a**), specificity (**b**), positive likelihood ratio (**c**), negative likelihood ratio (**d**), diagnostic odd ratio (**e**), and AUC (**f**) for bowel dilation diagnosis.

**Table 1 jcm-12-07714-t001:** Summary of main characteristics of included studies.

Article, Year	Country	Age Population	Type of Study	Study Population and Indication	N Male/Female	US Device	Contrast Agent	Amount of Contrast Agent	Ultrasound Diagnostic Criteria for CD	Gold Standard
Chatu, 2012 [11]	UK	36 (SD ± 15)	Retrospective	Suspected or already diagnosed CD. Performed for first diagnosis or detection of complications	64/79	Toshiba Medical Systems, Tochigi, Japan/GE Healthcare, Milwaukee, WI, USA	PEG Klean^®^ Prep (Norgine)	1 sachet in 1000 mL	(1) distended bowel wall thickness >3 mm; (2) absence of peristalsis; (3) stiffened bowel loop; (4) presence of a stricture, fistula, abscess; (5) absence of wall stratification; (6) increased power Doppler activity; (7) mesenteric lymph node hypertrophy; (8) mesenteric fat hypertrophy	Final diagnosis, enclosing CT
Hakim, 2019 [12]	UK	15 (2–17)	Retrospective	Suspected or already diagnosed CD. Performed for first diagnosis or detection of complications	49/44	Toshiba Medical Systems, Tochigi, Japan/GE Healthcare, Milwaukee, WI, USA	Pediatric PEG (Movicol)	1 sachet/250 mL 250–1000 mL (body weight-ba sed)	Same as [11]	Magnetic Resonance Enterography (MRE)
Kumar, 2015 [13]	UK	29.6 (SD ± 10.7)	Prospective	Patients with suspicion of CD. Performed for first diagnosis or detection of complications	12/13	Toshiba Medical Systems, Tochigi, Japan/GE Healthcare, Milwaukee, WI, USA	PEG Klean^®^ Prep (Norgine)	1 sachet in 1000 mL	Same as [11]	Magnetic Resonance Enterography (MRE)
Onali, 2012 [14]	Italy	44 (19–73)	Prospective	Patients with CD who needed surgery	8/7	Hitachi, EUB 6500, Japan	PEG (Promefarm, Milano, Italy)	375 mL (250–500 mL)	Same as [11], except lymphnodes enlargement (defined as >1 cm);	Surgical and pathological findings
Pallotta, 2012 [15]	Italy	37.7 (12–78)	Prospective	Patients with CD who needed surgery	28/21	Toshiba Tosbee (Tokyo, Japan)	PEG 4000 (Promefarm, Milan, Italy)	375 mL (250–500 mL)	Increased wall thickness (>3 mm). Bowel stenosis was defined as lumen diameter < 1. Bowel dilatation was defined as lumen diameter > 2.5 cm. Hypoechoic peri-intestinal lesions were defined fistulas when duct-like Abscesses in case of round-like mass with a diameter > 2 cm	Surgical and pathological findings

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
