# Peer review of "Small Intestinal Contrast Ultrasonography (SICUS) in Crohn’s Disease: Systematic Review and Meta-Analysis"

_jcm, 2023, doi:10.3390/jcm12247714_

Round 1
Reviewer 1 Report
Comments and Suggestions for Authors
Though this review ultimately contains a small number of studies and patients, I can appreciate the narrowness of the criteria in order to evaluate specifically small bowel disease. Were studies using ultrasonography without contrast excluded or unavailable? Further discussion of this would be warranted.
In part 3.2 "Final Diagnosis" is used a criteria but this is never defined. Is it withina certain time frame? Does it only include CD or any form of inflammatory bowel disease? A simple statement in the methods would clarify.
PEG powder is listed as being measured in mL but this should be a mass measure not a volume measure. The fluid would be a volume measure.
Would recommend the authors review the following work for more information in pediatrics:
Kellar, Amelia et al. “The Simple Pediatric Activity Ultrasound Score (SPAUSS) for the Accurate Detection of Pediatric Inflammatory Bowel Disease.” Journal of pediatric gastroenterology and nutrition vol. 69,1 (2019): e1-e6. doi:10.1097/MPG.0000000000002298
Kellar, Amelia et al. “Intestinal Ultrasound for the Pediatric Gastroenterologist: A Guide for Inflammatory Bowel Disease Monitoring in Children: Expert Consensus on Behalf of the International Bowel Ultrasound Group (IBUS) Pediatric Committee.” Journal of pediatric gastroenterology and nutrition vol. 76,2 (2023): 142-148. doi:10.1097/MPG.0000000000003649
Chavannes, Mallory et al. “Bedside Intestinal Ultrasound Predicts Disease Severity and the Disease Distribution of Pediatric Patients With Inflammatory Bowel Disease: A Pilot Cross-sectional Study.” Inflammatory bowel diseases, izad083. 25 May. 2023, doi:10.1093/ibd/izad083
Grammatical/Language:
Paragraph 1 Sentence 2: grammatical concern
Paragraph 2 Sentence 2: informal language, unclear placement of citation [2]
Paragraph 2 Last sentence: grammatical/language concern
The table is difficult to read due to spacing and disruption of words at the end of the lines. There are several missing spaces. Would recommend reformatting.
Row 6 column 1 of Table 1: the ultrasound diagnostic criteria wording is unclear.
The figures are too small to comfortably read.
Line 56: Language concern
Line 65: language concern
Line 81: language concern
Comments on the Quality of English LanguageOnly minor wording choices need review as listed above.
Author Response
Though this review ultimately contains a small number of studies and patients, I can appreciate the narrowness of the criteria in order to evaluate specifically small bowel disease. Were studies using ultrasonography without contrast excluded or unavailable? Further discussion of this would be warranted.
Our literature search did not find any study using ultrasound contrast meeting inclusion criteria. However, we discussed about the use of intravenous contrast as a possible future development for SICUS in the Discussion section.
In part 3.2 "Final Diagnosis" is used a criteria but this is never defined. Is it within a certain time frame? Does it only include CD or any form of inflammatory bowel disease? A simple statement in the methods would clarify.
The term “final diagnosis” connoted “the final judgement of the physician after performing all diagnostic tests” and refers only to CD. Unfortunately, a time frame was not described in the studies. We added this detail in the paper.
PEG powder is listed as being measured in mL but this should be a mass measure not a volume measure. The fluid would be a volume measure.
With the term “volume”, we indicated the total volume of the PEG solution. As the study clearly stated, we added in the table that one sachet of the branded powder was dissolved into the final water volume.
Would recommend the authors review the following work for more information in pediatrics:
Kellar, Amelia et al. “The Simple Pediatric Activity Ultrasound Score (SPAUSS) for the Accurate Detection of Pediatric Inflammatory Bowel Disease.” Journal of pediatric gastroenterology and nutrition vol. 69,1 (2019): e1-e6. doi:10.1097/MPG.0000000000002298
Kellar, Amelia et al. “Intestinal Ultrasound for the Pediatric Gastroenterologist: A Guide for Inflammatory Bowel Disease Monitoring in Children: Expert Consensus on Behalf of the International Bowel Ultrasound Group (IBUS) Pediatric Committee.” Journal of pediatric gastroenterology and nutrition vol. 76,2 (2023): 142-148. doi:10.1097/MPG.0000000000003649
Chavannes, Mallory et al. “Bedside Intestinal Ultrasound Predicts Disease Severity and the Disease Distribution of Pediatric Patients With Inflammatory Bowel Disease: A Pilot Cross-sectional Study.” Inflammatory bowel diseases, izad083. 25 May. 2023, doi:10.1093/ibd/izad083
Thank you for the suggestion. In the discussion, we reported and described the suggested studies in order to improve the framing of SICUS in pediatric population.
Grammatical/Language:
Paragraph 1 Sentence 2: grammatical concern
Paragraph 2 Sentence 2: informal language, unclear placement of citation [2]
Paragraph 2 Last sentence: grammatical/language concern
We corrected the sentences, as suggested.
The table is difficult to read due to spacing and disruption of words at the end of the lines. There are several missing spaces. Would recommend reformatting.
We changed the formatting of the table. We hope that it will be further improved in the final editing of the journal.
Row 6 column 1 of Table 1: the ultrasound diagnostic criteria wording is unclear.
We modified the text accordingly.
The figures are too small to comfortably read.
We have enlarged the size of figures
Line 56: Language concern
Line 65: language concern
Line 81: language concern
We rephrased the sentences in the cited lines
Reviewer 2 Report
Comments and Suggestions for Authors
This article is a meta-analysis of SICUS's utility in diagnosing Crohn's disease in adults and children. The analysis subjects and methods are acceptable. The results also show low sensitivity and high specificity for strictures and fistulae, prone to examiner experience bias. Conversely, the final diagnosis, abscesses, and dilatation, less prone to examiner bias, have extremely high sensitivity and slightly low specificity, which is typical of the results. Although the discussion is well-reviewed, it would be desirable to describe in more detail the limitation of examiner bias in the above five categories. In other words, the quality of the paper would be significantly improved if the literature search and citation of the results multiplied by gastrointestinal ultrasonography and examiner bias were shown to be consistent with the results of the above five categories. The paper is a touchstone for disseminating a new method, SICUS.
Author Response
This article is a meta-analysis of SICUS's utility in diagnosing Crohn's disease in adults and children. The analysis subjects and methods are acceptable. The results also show low sensitivity and high specificity for strictures and fistulae, prone to examiner experience bias. Conversely, the final diagnosis, abscesses, and dilatation, less prone to examiner bias, have extremely high sensitivity and slightly low specificity, which is typical of the results. Although the discussion is well-reviewed, it would be desirable to describe in more detail the limitation of examiner bias in the above five categories. In other words, the quality of the paper would be significantly improved if the literature search and citation of the results multiplied by gastrointestinal ultrasonography and examiner bias were shown to be consistent with the results of the above five categories. The paper is a touchstone for disseminating a new method, SICUS.
We thank the reviewer for the comment. In the Discussion paragraph we added a paragraph discussing this point, in particular by focusing on fistulae. The paragraph is the following: “The perspective is different for fistulae. The ultrasound pattern evocating a possible fistula may not be univocal and several possible signs have been proposed, often related to the experience of the examiner [21]. Indeed, some further evidence showed a sensitivity close to 70% for detecting fistulae, even when other ancillary techniques such as water immersion one were adopted [22]. However, results are conflicting in literature, as an additional study showed a sensitivity of 100% and a specificity of 98% [23], thus underlining that the expertise of the observer may be a main issue and a bias to be highlighted when reporting results in a systematic review. Rectum may be difficult to be examined by ultrasound, and this may explain why some cases of fistulae or abscesses may be missed by SICUS [24]. A different approach in this case, i. e. trans-perineal ultrasound examination, could add sensitivity to SICUS, as shown in some evidence from literature [25], provided that the operator has a sufficient level of expertise when scanning the perineal and perirectal areas.”